# Quantization-Guided Training for Compact TinyML Models

## ABSTRACT

We address the methodology to train and quantize deep neural networks (DNNs) in order to produce compact models while maintaining algorithmic accuracy. In this paper, we propose a Quantization Guided Training (QGT) method to guide DNN training towards optimized low-bit-precision targets, and reach extreme compression levels below 8-bit precision. Unlike standard quantization-aware training (QAT) approaches, QGT uses customized regularization to encourage weight values towards a distribution that maximizes accuracy while reducing quantization errors. We validate QGT using state-of-the-art model architectures (MobileNet, ResNet) on vision datasets. We also demonstrate the effectiveness with an 81KB tiny model for person detection down to 2-bit precision (representing 17.7x size reduction), while maintaining an accuracy drop of only 3% compared to a floating-point baseline.

## 1 INTRODUCTION

Deep neural networks (DNNs) have been at the core of many application breakthroughs [1–4]. They are transitioning from the cloud to the edge because of privacy issues, the need for real-time responses, and lack of network connectivity. One of the main challenges to enable efficient DNN inference is the ever-increasing number of parameters. Over the past decade, the number of DNN parameters has gone from millions to billions and is projected to reach 1 trillion parameters within the next decade.

To bring these applications (computer vision, natural language processing, and anomaly detection) to the edge and closer to the data source, we need to reduce the compute and memory footprints of DNN inference. Through quantization, parameters for DNN can be transformed into a lower bit-precision to support a smaller memory footprint and lower power consumption. For example, quantization algorithms can convert DNN parameters from 32-bit floating-point (FP32) to 8 or lower bit-precisions with minimal loss of accuracy.

This paper presents the *Quantization Guided Training (QGT)* method to tackle the problem of producing compact models while maximizing accuracy for a given model size. The term "guided" here refers to the training ability to adaptively nudge the model weights towards a more compression-tolerant optimum in the solution space of model parameters. Our philosophy is based on the notion that low-bit precision training represents training with additional dimensions presented by the bit-precision of the model parameters. Therefore, the DNN solution space can be significantly larger and

*TinyML 2021, March 2021, Burlingame,CA*

© 2020 Association for Computing Machinery.
ACM ISBN 978-x-xxxxx-xxxx-x/YY/MM...$15.00
https://doi.org/10.1145/nnnnnnn.nnnnnnn

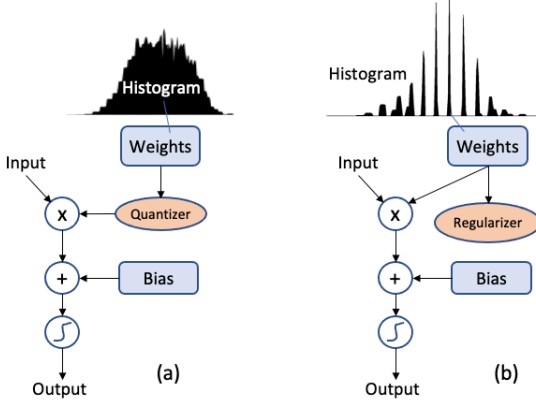

**Figure 1: Comparison of DNN training methods and optimized histogram of weight values. (a) standard quantization-aware training (QAT), (b) proposed quantization-guided training (QGT).**

more complex, and consequently, methods are needed to guide the search during training to arrive at a more near-optimal low-bit precision solution.

Fig. 1 shows a high-level diagram of our proposed QGT method compared to a traditional QAT method. Using regularizers attached to the model graphs, QGT helps nudge weight values closer to quantized bins to reduce quantization errors. Such a result in DNN solution space epitomizes QGT's ability to guide low-bit-precision training. Our approach is fundamentally different from current Quantization Aware Training (QAT) methods in that we directly influence DNN training through regularizer terms added to the loss terms. Rather than perturbing the training by weight approximation [5] or quantization noise [6], QGT influences the loss function by penalizing solutions that do not quantize well.

Fig. 1 also shows example histograms of weight values for a DNN layer. It is evident that the shape and distribution of the weight values are different between QAT and QGT approaches. QGT trained parameters are clustered around the bins defined by the respective regularizers. In this paper, we describe the QGT method to guide the DNN parameter values to these desired distributions.

The research area in DNN quantization, particularly in training methodology, is gaining momentum and moving at a very fast pace. To the best of our knowledge, we offer the following contributions in this paper:

- A novel training methodology using regularization terms added directly to DNN loss function to guide training towards low quantization error.
- Experimental results on state-of-the-art models on image classification tasks to demonstrate the effectiveness of QGT.

- A example use case for QGT for visual wakeup application, with results for tiny ML models at low bit-precision.

The rest of this paper is organized as follows: In Section 2 we discuss similar approaches and the motivation for our approach. In Section 3 we provide the details of the QGT method. Section 4 presents some results based on image classification tasks, including a visual wakeup case study using a person detection model. We describe the research impact of QGT, and finally, in Section 5, we present conclusions to our current work.

## 2 RELATED WORK

Many model compression methods deal with efficient parameterization, wherein model architectures are configured with a smaller number of parameters [7–9]. Similar methods to lower parameter count include fine-tuning steps such as pruning [10, 11], sparsity training [12] and weight sharing [13] that remove individual weights and reduce both memory footprint and inference time of the model. Other approaches include knowledge distillation [14] that trains a compressed model using a teacher-student pair. Other hybrid approaches include the *Lottery Ticket* method [15] that combines architecture search, sparsity training, and pruning to arrive at a compressed model.

Quantization, on the other hand, is a different compression method to lower the bit-precision for DNN parameters. Most quantization approaches are post-training quantization (PTQ) [5, 16, 17], where the DNN parameter values are assigned to quantized bins without re-training. Compared to quantization-aware training (QAT), PTQ is not as optimal because optimizations through training allow for a more comprehensive search of parameter values to achieve the best DNN accuracy. With QAT, the weights are quantized during training and the gradients are approximated with the straight-through estimator (STE). However, QAT approaches [5, 6, 18] requires explicit use of *fake-nodes* to the model graph to account for the rounding effects of quantization. Such an approach is laborious as it requires a considerable manual effort that is bespoke for each model.

Our proposed QGT method uses regularizers applied to the loss function to train using weight values, with the desired distribution and with low quantization errors. There is no dependence on the use of STE gradient approximation which may impact model convergence rate. Using regularizers, QGT can enforce properties such as clustering of weight values into quantized bins. QGT extends earlier efforts [19, 20] with refinements in training hyperparameters and regularizers to explicitly enforces weight values toward the desired sparsity and clustering targets.

## 3 QUANTIZATION GUIDED TRAINING

This section presents the detailed formulation of QGT. QGT, like other QAT approaches, is a tensor-level algorithm. It can be applied to all or any subset of the model parameters and can accommodate both fixed- and mixed-precision computational graphs. While we describe QGT in the context of symmetric and asymmetric quantization schemes for their prevalence and presentation clarity in this paper, we stress that QGT can accommodate other more specialized quantizers, including the powers-of-two quantizer [19, 20].

QGT is based on the premise that suitable model-parameter-based loss terms can be used as proxies for the quantized model performance. This is quite advantageous as it allows for retrofitting nearly any model-training pipeline into a QGT one with minimal overhead by adding these terms as regularizers. We refer to the proxy parameter-based losses used in QGT as *quantization-error losses*. Note that the term "error" here refers to the deviation of the model parameter from its dequantized version and should not be confused with the loss of the quantized model.

Since quantization-error terms are purely parameter-based, they can be regarded as regularizers. This is not just a semantic distinction and has an important implication. Regularizer-based approaches are computationally more economical for the reason that regularizers are computed only once for each batch, irrespective of the size of the batch. Another main advantage of QGT being a regularizer-based approach from the perspective of model training is that parameter-based loss functions have much more stable gradients. Therefore, even though quantization-error terms do not precisely capture the quantized model performance, by the virtue of producing more steady back-propagated gradients, they result in a more stable training than approaches that rely on the direct back-propagation of the quantized-model loss gradients. Standard QAT approaches using gradients approximated with straight-through estimator (STE) [5, 6, 18] are examples of training with quantized model loss. As such, model convergence using QAT may be susceptible to the variability of the model losses due to quantization, and potentially slower.

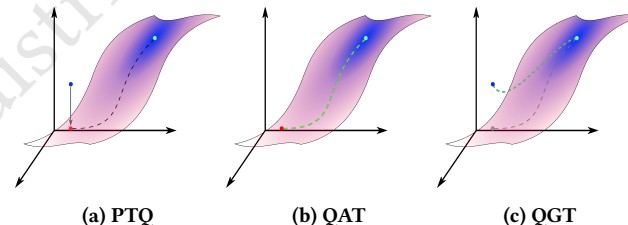

|         |         |         |
| :-----: | :-----: | :-----: |
| (a) PTQ | (b) QAT | (c) QGT |

Figure 2: Illustrations of (a) PTQ, (b) standard QAT and (c) QGT in model parameter space. Since the QGT loss is a combination of both task and quantization-error losses, the optimization trajectory can be guided towards an optimal solution without hard constraints.

Having discussed the main idea behind QGT at a high level, we next present the mathematical formulation of QGT. We start by showing why quantization-error terms are reasonable proxies for the quantized model loss. We use the symbol w to denote model parameters and $w_q$ for their dequantized counterparts. Mathematically, the quantizer, $Q$, is an operator that acts on w, and has the pseudo-inverse $\mathcal{D}$. This pseudo-inverse is, in fact, the dequantizer in the case of familiar quantizers. From this perspective, performance drop due to quantization can ultimately be attributed to the fact that $Q$ fails to be an invertible transform, i.e., $\mathcal{D}Q \neq \mathbb{I}$. The composite operator $\mathcal{D}Q$ maps w to its dequantized counterpart, $w_q$. As illustrated in Fig. 2, the set of tensors that coincide with their corresponding dequantized tensors form a subspace in the space of model parameter w. Within this subspace, the operator $\mathcal{D}Q$

becomes an identity operator, making the subspace a quantization-invariant subspace. The degree to which model parameters deviate from the quantization-invariant subspace correlates with the quantized model performance. QGT leverages this operator to obtain a distance from the quantization-invariant subspace.

Mathematically, it is most convenient to quantify this distance from the quantization-invariant subspace using the $L_2$ norm:

$$\mathcal{L}_{Q,\mathrm{w}} = ||\mathrm{w}_q - \mathrm{w}||^2, \tag{1}$$

where $\mathrm{w}_q = \mathcal{D}(\mathrm{q})$ with $\mathrm{q} = Q(\mathrm{w})$ being the quantized weight. Note that $\mathrm{w}_q$ is of the same type as w (i.e., FP32), whereas, the quantized tensor, q is the one in the desired representation such as 4-bit fixed-point. To directly relate this $L_2$ loss term to the quantizer, one can write:

$$\mathcal{L}_{Q,\mathrm{w}} = ||(\mathcal{D}Q - \mathbb{I})\mathrm{w}||^2. \tag{2}$$

QGT co-optimizes the loss terms $\mathcal{L}_{Q,\mathrm{w}}$ (one for each model parameter tensor) alongside the original model loss, $\mathcal{L}$, in the course of training. The full training loss under QGT is:

$$\mathcal{L}_{\mathrm{QGT}} = \mathcal{L} + \sum_i \lambda_i \mathcal{L}_{Q_i,\mathrm{w}_i}. \tag{3}$$

At the first glance, it may seem that QGT is just a Lagrange multipliers formulation of the same constrained optimization central to standard QAT approaches. The reason that this is not the case is that $\lambda_i$ parameters are not optimized and are treated as hyperparameters. Intuitively, they control how far the model can deviate from the quantization-invariant subspace when searching for a QGT-optimal solution. It is for this reason that we used the term co-optimization, and, as shown in Fig. 2c, why QGT is not a constrained optimization. In a sense, QGT offers a Pareto optimization search with $\lambda$ parameters furnishing a scalarization.

This formulation of QGT offers an important advantage: Since $\lambda_i$ parameters control the distance from the quantization-invariant subspace, it is possible to interpolate between an unconstrained training, which is effectively the same as PTQ as one always has to quantize at the end, and one akin to that of a standard QAT approach during training. This is illustrated in Fig. 3. When QGT's $\lambda$ parameters are small, QGT essentially reduced to PTQ. On the other hand, when $\lambda$ parameters are large, one effectively ends up with a hard QAT approach with constrained model space search. It is when $\lambda$ parameters are neither too small to be able to nudge model parameters, nor too large to overwhelm the task performance QGT becomes distinct from PTQ and QAT approaches. This flexibility is particularly useful when starting with a pre-trained model in cases where training is computationally too expensive or in situations where the model tends to get stuck when trained under a hard QAT approach. This also implies that QGT can be used as a fine-tuning augmented PTQ approach.

Let us see how QGT can alleviate some of the main shortcomings of PTQ and standard QAT approaches: The main issue with PTQ is that, as conveyed in Fig. 2a by the shading, there is no guarantee that the projection of the most task-optimal point onto the quantization-invariant subspace is also the most task-loss optimal point within the subspace. Thus, one may see significant improvement in performance with even a little fine tuning, which QGT

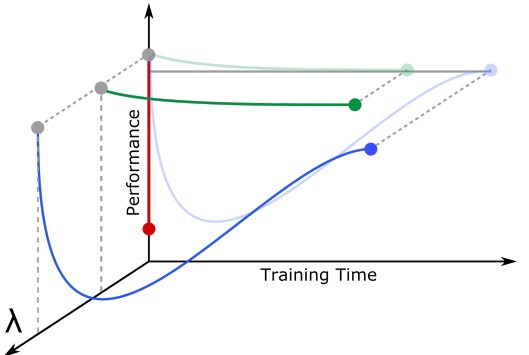

**Figure 3: QGT, depending on the $\lambda$ parameters, can serve as either PTQ ($\lambda = 0$ – the red curve) or a standard QAT ($\lambda \gg 1$ – the blue curve).**

offers. As for standard QAT approaches, constraining the optimization to the quantization-invariant subspace may take considerably longer to converge or, worse, get stuck. This issue becomes more pronounced at lower-bit representations where the quantization-invariant subspace shrinks significantly. Clearly, a less constrained optimization such as the one that QGT involves may find an off-subspace shortcut to the optimal point and evade getting stuck.

Any QAT approach without adequate training convergence produces a quantized model with sub-optimal performance, and QGT is no exception. The aspect that is specific to QGT, however, is that the quantized model performance is similar but not exactly the same as the model under training. The reason for this is that, as can be seen in Fig. 2c, it is only at convergence that the training trajectory in the model parameter space lands on the quantization-invariant subspace. This aspect allows for utilizing QGT not only as a versatile and flexible QAT approach, but also as an efficient fine-tuning-augmented PTQ technique by training for a few epochs.

## 4 RESULTS AND ANALYSIS

In this section, we present experimental results to evaluate the effectiveness of QGT on image detection tasks. We benchmark our results by comparing accuracy versus model size, with the goal to minimize overall accuracy loss. We first examine QGT from its ability to train models for standard benchmarks such as ImageNet and its variants. Then, we examine QGT from an application development for a visual wakeup system. Our design goal is to reach below 100KB in model size.

For evaluation, we used a workflow [21] that builds upon a TensorFlow-based framework to implement all of our training and quantization approaches. Once trained, we can then compile and generate either a native binary that can run independently or with a TFLite runtime. This workflow allows quick evaluation on embedded processor because the compiler targets optimal code for target hardware.

### 4.1 Benchmark Experiments

Table 1 presents the results of a number of QGT experiments for various architectures. Note that the main purpose of these experiments is to evaluate QGT's utility on small and large architectures in the

context of a range of tasks. We used the asymmetric quantizer in all of the experiments. We have specifically focused on four- and two-bit results as the performance of higher-bit post-train-quantized models (i.e., 8 and 16) are often close to that of their original floating-point models. In the experiments, the quantizer was applied, both, in per-tensor and per-channel (for convolutional and depth-wise convolutional layers) fashions. We do not apply QGT to biases and the trainable parameters ($\beta$ and $\gamma$ – see [22]) of the batch normalization layers. The reported top-1 accuracies were computed based on the dequantized weights with activations kept at four-byte floating point. Since the asymmetric quantization requires retaining slope and intercept, the sizes of per-channel-quantized models are slightly larger than their per-tensor counterparts quantized at the same bit-widths.

| Model | Accuracy (% top-1) | Δ Accuracy (% top-1) | Size (MB) | Δ Size Reduction | Bit Precision |
|---|---|---|---|---|---|
| MobileNetV1* ImageNet | 70.4 | - | 4.25 | - | FP32 |
| | **68.2** | -2.2 | 0.53 | 8x | 4 |
| ResNet50 ImageNet | 72.8 | - | 98 | - | FP32 |
| | **70.1** | -2.7 | 12.25 | 8x | 4 |
| MobileNetV1† ImageNette (grayscale) | 79.1 | - | 3.3 | - | FP32 |
| | **72.3** | -6.8 | 0.41 | 8x | 4 |
| | **77.3** | -1.8 | 0.45 | 7.3x | 4 p.c. |
| MobileNetV1† ImageNette (RGB) | 81.2 | - | 3.3 | - | FP32 |
| | **78.9** | -2.3 | 0.45 | 7.3x | 4 p.c. |
| | **69.5** | -11.7 | 0.25 | 13.2x | 2 p.c. |
| ResNet50‡ Eight Classes | 87 | - | 94 | - | 32 |
| | **84** | -3.0 | 11.75 | 8x | 4 |

**Table 1: Comparison of the FP32 and QGT-compressed performance and model sizes for a number of image classification tasks. The abbreviation "p.c." stands for per-channel. * $\alpha = 1.0$ and input size of $(224, 224)$. †† $\alpha = 0.5$ input size of $(128, 128)$. ‡ Classification on an eight-class subset of the COCO-2014 dataset (person, bicycle, car, motorcycle, airplane, bus, train, truck).**

The results in Table 1 suggest that QGT is able to effectively compress small and large DNN architectures regardless of the task / dataset complexity. We show that 4-bit bit-precision achieve significant compression (7-8×) while reducing only 1-3% drop in accuracy. We note that per-channel quantization provides higher accuracy than its per-tensor counterparts, at the same bit-precision targets. At 2-bit precision target, our early results are not conclusive as the solution space might require a more comprehensive search, or the model architecture might reach its capacity for the task / dataset.

To see why QGT, in spite of the simplicity of its formulation is so effective, it is instructive to take a closer compare the histograms of a model parameters trained under QGT against those of its base floating-point version. Fig. 4 presents several histograms (from the 27 convolutional kernels plus the final dense layer) of the MobileNetV1 model trained on the ten-class ImageNette task. The salmon-color histogram is the 4-bit per-tensor asymmetric dequantized trained model under QGT, and the pale blue histograms

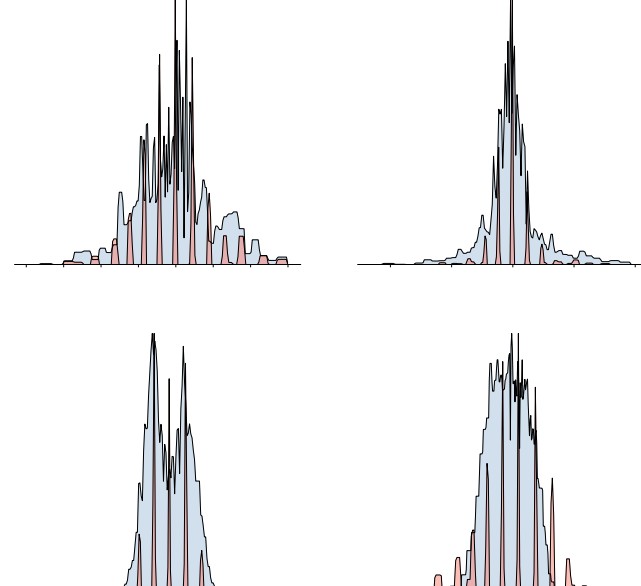

**Figure 4: Comparison of kernel histograms of MobileNet V1 with $\alpha = 0.5$ trained on the 10-class ImageNette dataset with (salmon) and without QGT (pale blue) at convergence. The bottom left histogram is the kernel of the dense layer.**

is that of the base floating point model. We make a number of important observations: First and perhaps most obviously, since the model trained under 4-bit asymmetric per-tensor QGT, we end up with the binning of the dequantized weight tensor entries into $2^4$ bins. Second, in most cases, it appears that the trained model under QGT converges to distributions that closely resemble their base-model counterparts. This seems to be the case when the weight histogram shape retains the bell shape of its initialization. This, perhaps, can be attributed to the fact that such tensors lack a subset of entries with oversized relevance to inference entailing large gradients that bring about homogeneous distortion. Third and related to the earlier point, the dequantized weight tensor histogram deviates significantly when the floating-point weight histogram is rather too distorted compared to its originally initialized distribution. These observations further bolster the flexibility and effectiveness of QGT.

## 4.2 Wakeup Systems

Wakeup systems are good example applications that can best benefit from QGT trained models. With a low-cost compute platform, savings in memory footprint becomes important to support the limited on-chip memory and processing capability. Furthermore, wakeup systems are always on, and as such, low power consumption and low false alarms are critical requirements. For this paper, we focus on a computer vision use case of identifying whether a person is present in the image or not. Applications for such a person detection model include surveillance/security in entryways and passenger detection for in-vehicle use.

In our study, we focus on the MobileNet (V1 and V2) architecture. Person detection models for visual wakeup systems have been previously reported at 8-bit precision at 208KB using MobilenetV1 with depth multiplier 0.25, and 290KB using MobilenetV2 with depth multiplier 0.35 [23]. Typical edge processors / microcontrollers for wakeup systems have extremely limited on-chip memory (100-320KB SRAM) and flash storage (up to 1MB). The DNN model parameters and associated inference code have to fit in memory buffer, with sufficient allocation for buffers for input/output data. In this paper, we push the envelope further to achieve a smaller DNN footprint to support additional models that can be processed concurrently.

To train the MobileNet models using QGT, we leverage the COCO-2014 dataset for sample labeled images for person and non-person categories, much like the Visual Wakeup dataset [23]. Fig. 5 show example images for small and large pixel on target person object (0.5% and 10% of the image, respectively). We evaluated the dataset for images with a minimal of approximate 100 vertical pixels on the person object, which is roughly 10% of the VGA sized images in COCO-2014. This is done to match our camera resolution (160×120) at the anticipated target distance to objects in our application use. Specifically, when the VGA sized image is resized to our camera resolution during training, we need to make sure there is physically enough texture and shape information.

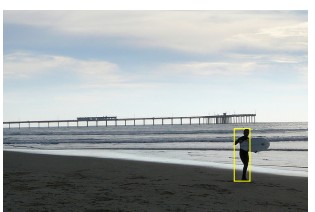 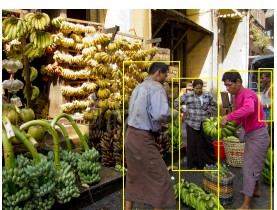

**(a)** Small person object below 0.5% of image **(b)** Large person objects above 10% of image

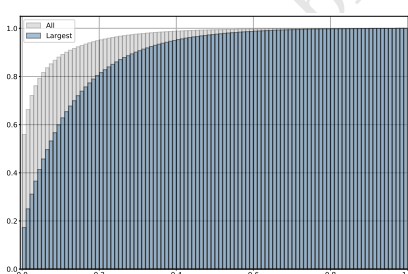

**(c)** Histogram person objects sizes - normalized area, largest instance (blue) and all instances (gray) containing person

**Figure 5: Sample COCO-2014 images for (a) small and (b) large person size, with normalized distribution in dataset (123,287 total, 66,608 with at least one person).**

As shown in Fig. 5c, over 60% of the positive images contain a largest instance of a person that is smaller than 10% of the image. With a size threshold of 10%, over 60% of the images containing a person have to be excluded. From this, we train our MobileNet models with assigned labels (person / non-person), while addressing class imbalance [24] between the two classes by undersampling the

non-person images and adjusting the operating thresholds of the final classification layer.

We evaluate the person detection models trained with QGT with different target bit precision to demonstrate the effectiveness of training methodology. Our goal is show the size reduction at low bit-precision, with minimal drop in accuracy. We selected the MobileNet (V1 and V2) architecture, with 0.25 depth multiplier for comparison purposes.

We first established a baseline floating point (FP32) model trained to convergence. We use PTQ to quantize the baseline FP32 model to 4bit and 2bit precision. Then we train the models with a sweep of QGT lambda hyperparameter (see Section 3), also with 4-bit and 2-bit target. We explored both per-tensor and per-channel quantization schemes with QGT. Since per-channel provided better compression ratio and for clarity-sake, we show only per-channel results. In our experiments, we found that bias tensors are more sensitive to quantization, and as such, we left bias tensors untouched. Since bias tensors are small with respect to number of tensor elements, the overall impact to memory footprint is negligible. Finally, we do fold in the batch-norm layer to evaluate the model accuracy and memory size. The sizes are calculated based on packing weight values, since there are no standard formats for sub 8-bit models.

Table 2 shows the accuracy and size comparisons for the person detection model. We note that the accuracy levels at 4bits and 2bits (90.3% and 87.3%, respectively) for MobileNetV2 are only within 0.1 and 3% drop from the FP32 baseline. When comparing to PTQ results at 1% and 22% drop, respectively, we find that QGT can help maintain accuracy with training for lower bit-precision. Similar general trend is observed for MobileNetV1. As a reference, earlier result from [23] on person detection model was 85% at 250KB size with 8-bit precision.

| Model | Accuracy (%) | Δ Accuracy (%) | Size (KB) | Δ Size Reduction | Method, Precision |
|---|---|---|---|---|---|
| **MobileNet V1** ($\alpha = 0.25$) | 88.7 | - | 834 | - | FP32 |
| | 72.2 | -16.5 | 123 | 6x | PTQ, 4bit |
| | 52.7 | -36.0 | 73 | 11.4x | PTQ, 2bit |
| | **87.9** | -0.8 | 123 | 6x | QGT, 4bit |
| | **82.0** | -6.7 | 73 | 11.4x | QGT, 2bit |
| **MobileNet V2** ($\alpha = 0.25$) | 90.4 | - | 1440 | - | FP32 |
| | 89.1 | -1.3 | 133 | 10.8x | PTQ, 4bit |
| | 68.4 | -22.0 | 81 | 17.7x | PTQ, 2bit |
| | **90.3** | -0.1 | 133 | 10.8x | QGT, 4bit |
| | **87.3** | -3.1 | 81 | 17.7x | QGT, 2bit |

**Table 2: Quantization results for person detector tiny models, showing superior QGT results over PTQ and baseline for accuracy and size.**

From a memory size perspective, QGT offers 17× smaller footprint compared to the FP32 baseline, and a 3× compression compared to the aforementioned 8-bit results [23]. From a processing latency perspective, we are working with a number of hardware accelerators / SoC that supports sub 8bit processing (proprietary

info). As a reference for this paper, on ARM A72 processor (Raspberry Pi4), we measured approximately 246 fps (frames per second) for the MobileNet V1 model. We ran on 8-bit configuration because the A72 processor does not have sub 8-bit acceleration support. In general, we see a ~5× latency improvements with 8-bit versus FP32 inferences.

### 4.3 Analysis

In this subsection, we offer additional insights on QGT based on our DNN training experiences and results.

**Impact of learning rate**. We evaluate QGT over DNNs with increasing depth and number of parameters. Our training results show the QGT's $\lambda$ hyperparameter that governs *quantization-error losses* acts as an dynamic learning rate that is dependent to bit-precision. As such, QGT regulates learning in tandem with the global learning rate hyperparameter. One interpretation is that bit-precision is coupled with the learning capacity of the DNN (i.e., higher bit-precision can afford higher learning capacity, but at the cost higher memory footprint). In a sense, QGT regularizers is related to dynamic learning rate schedulers such as AdaGrad [25]. When we train with QGT, we can govern QGT's $\lambda$ and global learning rate hyperparameters to guide training towards an optimal solution space. That is, we can choose to train slowly to closely converge, or we can choose higher learning rate to move fast through the model parameter space. Future work will further explore the theoretical underpinnings of QGT regularization, with evaluations on larger datasets and deeper models.

**Impact of training time**. Most published research on model compression do not point out convergence speed. Admittedly, this aspect is very much depended on task and model complexity. Training time is also a function of available hardware allocated for training. Newer studies such as [26, 27] are starting to consider training time as a part of the greener AI efforts to reduce carbon footprint. With QGT, we have the additional $\lambda$ hyperparameters that, in theory, can better reduce training time and guarantee convergence (see Section 3) with a guided search. Future work will include a more comprehensive study on training time.

**Impact on model size and capacity**. Limited on-chip memory will be a major constraint in deploying DNN models such as MobileNet on constrained edge processors. Results in our study using QGT to train and quantized models show that deployment of tiny vision models are possible, with sizes well below 100KB. To reach such model sizes at such low bit-precision, guided training approaches are important tools for the tiny ML community. Future studies can address model capacity to further explore hybrid bit-precision amongst the layers of the DNN.

## 5 CONCLUSION

We show that guided training maintains performance in high quantization regime. We validated our proposed Quantization Guided Training (QGT) works with a variety of deep neural networks and datasets, using a number of quantization schemes. Our method can be applied to hybrid methods with mixed bit-precision to achieve extreme compression ratio at low bit-precision. QGT imposes a soft DNN training constraint and can be used with other training-aware approaches, e.g. QAT (quantization aware training) and PTQ (post training quantization), and weight pruning. We also demonstrate the effectiveness with QGT trained model at 2-bit precision for visual wakeup application.

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
