# OpenReview forum: "Quantization-Guided Training for Compact TinyML Models"
_tinyml.org/tinyML/2021/Research_Symposium — tinyML 2021 Regular_

### Official Review · AnonReviewer4 · 2021-01-29

**Overall Merit Score:** 2

**Brief Summary:**

In this paper, the authors propose quantization-guided training (QGT) which is better than the existing quantization-aware training (QAT) and post-training quantization (PTQ). QGT adds a quantization-related regularization term on the loss function, in order to guide the training to automatically find a quantized model.


**Detailed Comments:**

1. Including a quantization-related regularization term in the loss function is not a brand new approach. As examples (there are also other works):
Learning Low Precision Deep Neural Networks through Regularization (Samsung)
SinReQ: Generalized Sinusoidal Regularization for Low-Bitwidth Deep Quantized Training (ICML 2019 Workshop)

2. The paper needs to cite some additional previous work.
Though some QAT conduct optimization on the quantization-invariant space, there are works that use incremental quantization or layerwise quantization-aware fine-tuning. That way the optimization can also take a shortcut as QGT does.
For example:
Incremental network quantization: Towards lossless cnns with low-precision weights. (ICLR 2017)
Hawq: Hessian aware quantization of neural networks with mixed-precision (ICCV 2019)

3. Quantizing ResNet50 on ImageNet is an important benchmark. However, the ResNet50 pre-trained model in the paper has a 72.8% top-1 accuracy, which is significantly lower than normal ResNet50 that has >76% accuracy. In addition, the quantization performance (-2.7% with 4bit) may not beat some previous works, additional comparisons are required.

4. The advantage of QGT is faster convergence, this requires ablation study against general QAT methods.


**Paper Strengths:**

- The illustrations in the paper showing that QAT and PTQ are extreme cases of QGT are good. The way to consider QAT as an optimization with quantization constraints is clear.
- In this paper the authors try many different tasks with different complexity and targets, which is helpful to justify the proposed method.

**Paper Weaknesses:**

The novelty of this paper is limited. And the paper didn’t include some important related work. The experiments in this paper lack comparison with previous methods. An ablation study to justify the effectiveness of QGT on quick convergence is required.

**Poster (If Paper Is Rejected):**

1: Yes, ok for poster sesion to nurture work

**Reviewer Confidence:**

5: The reviewer is absolutely certain that the evaluation is correct and very familiar with the relevant literature

---

### Official Review · AnonReviewer1 · 2021-01-29

**Overall Merit Score:** 2

**Brief Summary:**

This paper studies the training of quantized neural networks. The authors introduce Quantization- Guided Training (QGT) that guides DNN training towards optimized low-bit-precision targets. Different from the conventional quantization-aware training (QAT), their proposed QGT uses a customized regularization to encourage weight values to be closer to their quantized counterparts.


**Detailed Comments:**

My major concern of this paper is on its limited technical novelty and insufficient experimental evaluations. Based on this, I think the paper is not ready for publication in its current shape.

**Paper Strengths:**

The topic of this paper is relevant to the theme of the conference. The authors have conducted some experiments in the TinyML setting, where they have achieved relatively good performance. The paper is also well-written and easy to follow.


**Paper Weaknesses:**

The technical contribution of this paper is rather limited. The idea of applying regularization to the quantization-aware training is not very novel as it has been widely studied in the literature: e.g., L2 and sawtooth regularization [Tang et al., AAAI 2017], Gaussian mixture regularization [Georges et al., INTERSPEECH 2019], and L1 gradient regularization [Alizadeh et al., ICLR 2020]. The authors should mention and discuss them in the related work section and provide a quantitative comparison with them to verify the effectiveness of their proposed method.
The experimental evaluations in this paper are not very solid. The only baseline that the authors have provided is the post-training quantization (PTQ) which is only evaluated on the wakeup systems. On the widely-adopted ImageNet benchmark, the authors have only provided the performance of their proposed solution. The authors should at least provide the performance of the conventional quantization-aware training (QAT) on both settings to verify their claim that QGT performs better than QAT. It is also necessary to include the results of other regularization-based quantization.
References:
[1] Tang et al., "How to Train a Compact Binary Neural Network with High Accuracy?", AAAI 2017.
[2] Georges et al., "Ultra-Compact NLU: Neuronal Network Binarization as Regularization", INTERSPEECH 2019.
[3] Alizadeh et al., "Gradient L1 Regularization for Quantization Robustness", ICLR 2020.


**Poster (If Paper Is Rejected):**

1: No, paper is below bar for poster as well

**Reviewer Confidence:**

5: The reviewer is absolutely certain that the evaluation is correct and very familiar with the relevant literature

---

### Official Review · AnonReviewer3 · 2021-01-29

**Overall Merit Score:** 2

**Brief Summary:**

The paper provides a regularization-based method called
Quantization Guided Training (QGT), to train quantized DNNs.
Unlike conventional quantization-aware training (QAT) approaches,
QGT uses customized regularization to encourage weight values
towards a distribution that maximizes accuracy while reducing
quantization errors.  QGT is validated using MobileNet and ResNet
on computer vision datasets. QGT is also validated using an 81KB tiny model
for person detection with 2-bit precision while maintaining an
accuracy drop of only 3% compared to a floating-point baseline.


**Detailed Comments:**

The proposed regularization-based QGT could be a good idea due to
its simplified procedure in training a quantized model, provided that the achieved accuracy performance is competitive. But the
paper does not compare its performance against the conventional
QAT. Based on the paper, it is unclear that the proposed QGT is competitive in
achieved accuracy compared to QAT.


**Paper Strengths:**

The proposed regularization-based method appears to be novel, and
the method streamlines model training for quantized networks.


**Paper Weaknesses:**

Since the paper proposes the regularization-based QGT method as
an improvement over QAT (Figure 1), QGT must be compared against
QAT.  Unfortunately, the paper does not provide a performance
comparison between QGT and QAT.

It is unclear from the paper if conventional quantization is
applied at the end of the QGT training phase to ensure that each
weight is set to a 4-bit fixed-point value, or if they are
tightly clustered around these points due to the proposed
regularization. If it is the latter case, then the approach
proposed in this paper would be interesting, but this reviewer
cannot tell.

**Poster (If Paper Is Rejected):**

1: Yes, ok for poster sesion to nurture work

**Reviewer Confidence:**

4: The reviewer is confident but not absolutely certain that the evaluation is correct

---

### Official Review · AnonReviewer2 · 2021-01-30

**Overall Merit Score:** 4

**Brief Summary:**

This paper presents a new approach to training for quantized model deployment, that uses loss functions to guide the weights towards appropriate values. The authors present results across a variety of computer vision tasks.

**Detailed Comments:**

Very promising approach, detailed comments inline above.

**Paper Strengths:**

The strengths, with notes, are listed below:

### Loss functions to guide quantized training

The key advance in this paper is the use of error loss functions to help regularize the weights and guide them towards values appropriate for a minimal accuracy loss during quantized inference. From my knowledge of the field this is a unique approach, and points to further interesting directions to explore in the future.

### High accuracy with two-bit weights

The results on the visual wakewords task using only two bit weights demonstrate an impressively high level of accuracy compared to the state of the art.

**Paper Weaknesses:**

### Lack of eight-bit comparisons

The key tables (#1 and #2) only show results for 32-bit floating point and 4-bit or 2-bit quantized inference. Since 8-bit inference is the current standard approach for quantization, it makes it hard to compare how well this approach compares to post-training quantization (though there is a mention of the 85% accuracy for 8-bit VWW in the preamble to the table, which helps).

### Few tasks

Only results for ImageNet and Visual Wakewords are included. It would be helpful to understand if this approach holds for a wider variety of benchmarks, especially audio, since those are important for embedded applications.

**Poster (If Paper Is Rejected):**

1: Yes, ok for poster sesion to nurture work

**Reviewer Confidence:**

5: The reviewer is absolutely certain that the evaluation is correct and very familiar with the relevant literature

---

### Decision · Program_Chairs · 2021-02-05

**Decision:**

Accept (Regular)

**Comment:**

Congratulations on your paper's acceptance!

Your paper has been accepted as a full-length regular paper.

Please read the reviews carefully and make sure the concerns are addressed in your final submission.

All accepted papers will be given a slot in the TinyML Summit schedule for an oral presentation on Friday, March 26, 2021.

Camera ready instructions will follow soon. All papers will be hosted on arXiv and published papers will have the following header stamp: “Published as a conference paper at TinyML Research Symposium 2021.” The paper will also be presented on the program website.